# Does the Synergy Model Implementation Improve the Transition from In-Hospital to Primary Care? The Experience from an Italian Cardiac Surgery Unit, Perspectives, and Future Implications

**DOI:** 10.3390/ijerph19095624

**Published:** 2022-05-05

**Authors:** Federica Dellafiore, Rosario Caruso, Tiziana Nania, Francesco Pittella, Tiziana Fiorini, Maria Paola Caruso, Giovanni Zaffino, Alessandro Stievano, Cristina Arrigoni

**Affiliations:** 1Department of Public Health, Experimental and Forensic Medicine, Section of Hygiene, University of Pavia, 27100 Pavia, Italy; federica.dellafiore@unipv.it (F.D.); cristina.arrigoni@unipv.it (C.A.); 2Health Professions Research and Development Unit, IRCCS Policlinico San Donato, 20097 San Donato Milanese, Italy; rosario.caruso@grupposandonato.it (R.C.); francesco.pittella@grupposandonato.it (F.P.); tiziana.fiorini@grupposandonato.it (T.F.); mariapaola.caruso@grupposandonato.it (M.P.C.); giovanni.zaffino@grupposandonato.it (G.Z.); 3Department of Biomedical Sciences for Health, University of Milan, 20133 Milan, Italy; 4Centre of Excellence for Nursing Scholarship, OPI, 00192 Rome, Italy; alessandro.stievano@gmail.com

**Keywords:** care, discursive paper, organizational, patient-centered model, primary care, healthcare system, Synergy Model

## Abstract

The demand for care services in the healthcare system has changed and is triggering a smooth transition from in-hospital to primary care. In this regard, patient-centered-care models of care delivery might provide a framework to follow patients’ journeys throughout their transition between different levels of care. Accordingly, an Italian research group at a cardiac hospital in Northern Italy implemented the Synergy Model in a Cardiac Surgery Unit, a patient-centered-care model, and is using the framework of the model to guide a smooth transition of patients towards rehabilitation and primary care after their hospitalization. This discursive paper is focused on the experience, perspectives, and future implications of adopting the Synergy Model to facilitate the transition from in-hospital to primary care. The presented experience and discussion might be helpful to the international debate regarding the strategies to boost a smooth transition from in-hospital to primary care.

## 1. Introduction

The personalization of nursing care, in response to the characteristics of patient needs and the evolution of the concept of health, is becoming increasingly compelling and challenging. This is especially true considering that patients often have to experience passages across different levels of care, from primary care to tertiary care, and vice versa. While the population becomes older than in previous decades, care services have become more multi-factorial as a response to chronic disease related conditions. Overall, patients are more educated, informed, and conscious about their rights and health conditions, and the expectations placed on national healthcare systems have increased. Therefore, the demand for care services in every healthcare system in high-income countries has profoundly changed in the last few years [1]. In this context, the concepts of efficiency, efficacy, and appropriateness play key roles in the attempt made by healthcare services to equally distribute resources and services, acknowledging the constant lack of available resources for welfare and health [2].

Therefore, implementing organizational delivery models of personalized nursing care is pivotal to optimizing the best utilization of the available resources to personalize care and determine a smooth transition across different levels of care. Personalized nursing care models, theoretically, ensure a patient-centered approach and address the care delivery in practice [3], enhancing nursing competencies, and consequently, reducing the gap between what is desired and what is experienced in daily practice [4]. The models mentioned above, also called patient-centered care (PCC), aim to take care of the patient from a holistic perspective. They include the patient’s family, social status, and all the features that make them patient-assisted, “unique” individuals [3]. The term patient-centered has been used since 1969 to highlight the need to consider assisted patients based on their uniqueness. The application of this term is parallel with the development of health psychology, as seen in the works of the Hungarian psychoanalyst Michael Balint, and Peschel e Peschel (1994) in the mid-1990s. In other words, PCC is based on a new approach built on communication, education, and empathic support for the assisted patients and their families [5].

Therefore, it is crucial in a patient-centered healthcare path to identify the care needs, connecting practice with scientific evidence and patient aspirations, desires, and wills. Moreover, PCC implementation models require shared decision making involving several professionals. This is because the nursing staff is the linchpin of connecting several specialties and professionals, within a structured ecosystem, for facilitating real, shared decision making. PCC requires an organizational strategy to improve modern healthcare systems, especially considering that patients have to use services belonging to different levels of care during the traditional pathway of several chronic conditions [5]. The current scientific evidence recognizes positive outcomes of PCC models in patients, healthcare personnel, the healthcare structure, and spending [6]. These models are related to better outcomes for people that are assisted, greater autonomy and coordination in the nursing care delivery, better efficacy and efficiency in the health care procedures, increased staff satisfaction, and reduced healthcare costs [7,8,9]. This discursive paper is focused on the experience, perspectives, and future implications of adopting the Synergy Model to facilitate the transition from in-hospital to primary care.

## 2. The AACN Synergy Model for Patient Care

One of the PCC models is the AACN Synergy Model for Patient Care (Synergy Model), developed by Dr. Martha Curley in 1998, within the activities of the American Association of Critical-Care Nurses (ACNN) recommendation [10]. The original purpose of the model was to provide a theoretical framework for certified practice [10,11,12,13]. Its major innovation introduced was that the patient outcomes were optimized when patient characteristics matched nursing competencies [14,15,16]. A pillar of the Synergy Model is that nursing competencies are guided and directed by the assisted patient’s characteristics [17,18,19,20]: when patient characteristics are in synergy with the nursing competencies, a “therapeutic relationship” is possible and can increase the chances for better clinical and welfare outcomes [10,15,16,21,22,23,24]. Patients and their families, and the ward characteristics, are unique. Considering that the model considers this “uniqueness”, its implementation is suitable for various contexts. Patient characteristics might also describe their transition between “health” and “disease” when relevant outcomes are considered as patient characteristics.

Accordingly, the Synergy Model identifies eight patient characteristics and eight nurse competencies [16,21]. These characteristics are resiliency, vulnerability, stability, complexity, resource availability, participation in care, decision making, and predictability. When a patient is evaluated following the Synergy Model, each of the eight characteristics is evaluated by employing a numeric value to rate the assessment: 1 (very low), 3 (moderate), or 5 (high). For example, a patient evaluated for resiliency, or the ability to return to the baseline level of functioning after an illness or injury, is rated a level 1 if the patient had a very low level of resiliency, 3 for moderate resiliency, and 5 for a high level of resiliency. Each patient characteristic is evaluated similarly.

Similar to the evaluation of a given patient, the nursing staff is evaluated on a scale from 1 to 5, based on evaluations from competent (1) to expert (5). In fact, the Synergy Model was conceived in the 1990s as a conceptual framework for certification. Its use has grown in numerous areas, such as exam certifications, university nursing programs, and as a reference point for professional practice in the hospital. The AACN has conceived a model to deliver and organize healthcare delivery guided by the needs of patients and their families, where the activities of nurses answering to patients’ needs are well-highlighted [13,14]. This model is based on the synergy between patients’ needs (including their families) and the nursing competencies [10,16]. The Synergy Model delineates the following eight competencies as the essence of professional nursing practice: clinical judgment, clinical inquiry, caring practices, response to diversity, advocacy and moral agency, facilitation of learning, collaboration, and systems thinking. Therefore, the synergy is given by the balance between patient characteristics and nursing competencies. Historically, the Synergy Model for Patient Care has been used in various settings, from inpatient care to military contexts [19,25,26,27].

Research carried out in American hospitals showed that the Synergy Model seemed to positively impact nurses’ satisfaction and patient clinical outcomes (e.g., better management of illness and symptoms, achievement of a good self-care level, reduction in terms of rehospitalization, feeling of involvement in the healthcare process) [28]. Likely, at organizational levels, the Synergy Model showed to be associated with positive outcomes (e.g., savings, turnover decrease, lower absenteeism, increased professional satisfaction, better management of human resources) [28]. Additionally, structures choosing the Synergy Model have greater possibilities to develop requirements for the “Magnet Hospital” certification [29]. This certification is conferred to the hospitals that provide a healthy work environment, where nursing professional growth has been enhanced, and where excellent clinical results have been proved [30].

Despite the Synergy Model and similar PCC, approaches are becoming more effective in responding to the real needs of patients and are having a great impact on people’s health. However, its implementation process is still challenging, and there is no application of this model in Italian clinical practice. Several empirical studies have highlighted the benefits of implementing the Synergy Model in various healthcare settings with different patient populations, supporting the rationale that adopting the Synergy Model might provide a guide for nurses working in tertiary care (in-hospital settings), secondary care (outpatient settings), and primary care (community level) [29,31,32,33,34].

## 3. Adaptation of Synergy Model to Cardiac Surgery Unit: An Applicative Experience

IRCCS Policlinico San Donato (PSD) has been deeply interested in individualizing nursing assistance over the last few years. For that reason, PSD shared the project to implement the Synergy Model (still in progress) inside its structure. In this paper, we describe our experience regarding the adaptation of the Synergy Model in a Cardiac Surgery Unit and the guidance of a smooth transition of patients towards rehabilitation and primary care services.

Initially, the implementation of the model required the development of a system to map nursing competencies and education, monitoring the nursing delivery process following the Synergy Model. The hypothesis is that in the settings that implement the Synergy Model, patient outcomes (e.g., gratification, engagement in the care process, safety, and clinical outcomes) and workers’ outcomes (e.g., better organizational health perception, increased self-efficacy, and more organizational well-being related outcomes) might improve [35]. This project started at the beginning of 2015 and continued until 2021 and consisted of different steps. Each step is dynamic and was developed in overlying periods without strict passages from one period to another.

All the phases have been divided following simplicity and clearness criteria:-Phase 1: Analysis of perceived barriers by PSD nursing staff to the implementation of the Synergy Model (started in 2015; ended in 2016);-Phase 2: Organizational improvements within the PSD organization to fill the gaps (barriers) with interventions to update clinical documentation, competency evaluations, and learning curves of nurses (started in 2017; ended in 2018), and implementation of the Synergy Model at the Cardiac Surgery Unit and supporting the transition of patients towards rehabilitation and primary care services (2019–ongoing).

## 4. Analysis of Hindrances Underpinning the Implementation of the Synergy Model

The first phase of the project was to examine and interpret the barriers perceived by the nursing staff as obstacles to the following Synergy Model implementation through a mixed sequential method study [36]. Phase 1 explored the feelings and personal perceptions of the nursing staff in relation to the barriers perceived as obstacles to the implementation of a new patient-centered care organizational model (i.e., the Synergy Model) using a qualitative approach. Phase 2, using a quantitative, observational, and cross-sectional design, described on a large scale the main intra-professional barriers concerning the problem under examination, through the administration of a questionnaire built ad hoc based on the results of Phase 1 [36]. Phase 1 was conducted between 2015 and 2016 and consisted of semi-structured interviews following a grid of five predefined questions to the nursing staff. Ahead of the interview, it was necessary to hold two editions of the continuing medical education (CME) course at PSD (May 2015). CME courses explained the Synergy Model and its potential applications, in the hospital mentioned above, to head nurses and selected clinical nurses. At the end of the course, nurses and head nurses were selected for the interviews voluntarily. The inclusion criteria required nurses to (a) be clinical nurses or coordinators with (b) seniority of more than five years. The data collection took place between June and July 2015, with a dynamic–purposeful sampling among nurses trained on the model, which invited them to participate in the study. The interviewer was previously subjected to training through role playing with a researcher experienced in managing interviews. The semi-structured interviews were audio-recorded, with the consent of the participants, and transcribed verbatim. After reading and rereading the interviews, a textual content analysis was performed to investigate the text in-depth and identify the issues that may have remained hidden from a superficial reading. The credibility analysis was carried out dynamically during the thematic analysis and planned to clarify any conflict of meaning during the synthesis and coding. The scientific validity of the entire qualitative research process was guaranteed by also using the methodology of checking between members of the research team and verifying that the proposed analysis was in line with the interviewees’ thinking (member-checking). Eight underlying themes were identified after the analysis of the data coming from the semi-structured interviews (deductive stage): (a) lack of inter-professional relationships, (b) standing-in, (c) administrative/bureaucratic overload, (d) sterile working days, (e) unsuccessful projects, (f) clinical vulnerabilities, (g) cultural weaknesses, and (h) organizational failings. Afterwards, in the abductive second reading of the first eight underlying themes, three main themes emerged and described the daily discomfort nurses had in their working conditions: (a) improper working, (b) ineffective working activities, and (c) limited added value.

Given the results coming from the interviews, it was possible to provide a questionnaire to 117 nurses in order to define perceived hindrances from a quantitative approach (Phase 2). Inside the questionnaire, the most relevant hindrance domain was related to professional growth, followed by working for an organization, and lastly, the nursing competence domain [36]. Therefore, it was possible to highlight and understand from an intra-professional nursing perspective the main barriers that prevented the Synergy Model’s implementation at PSD. It is crucial to grade the motivational obstacles to the implementation with a bottom-up viewpoint to improve hospital organizational health. Understanding the motivational hindrances is part of the logic of realizing an organizational healthcare knowledge that is keen on individualizing the cure, and it paves the way for a concrete basis for a real synergy project.

## 5. Organizational Changes Inside PSD According to Synergy Model Features

In this phase, conducted between 2017 and 2018, it was possible to identify the strategy according to Synergy Model features. Hence, it consisted of different organizational changes, such as applying standardized interventions, organization of clinical documentation (to standardize assessments and communication across staff members), and nursing and support staff’s education. Specifically, this method helped create a working team with distinct educational backgrounds and competencies regarding the Synergy Model. This group, called Synergy Group, was made up of one Health Management nurse, two head nurses from the Critical Care Unit, two from the Health Professions Research and Development Unit, a nurse operating in the Nursing Degree Course, and a referent nurse from the business training. All interventions and projects that were applied are described below:(a)Every three months, Synergy Group provided the Health Management, Chief Nursing Officer, and Chief Executive Officer an updated relation of the project status, explaining all the activities and discussing the improvements of implementation that had been carried out, involving several experts such as, nurses, supporting staff, and head nurses and physicians in the Operative Units.(b)Mapping the skills of nursing staff: Synergy Group developed a survey to stratify the nursing staff and identify their demographical features, under-graduation and post-graduation education, professional field, years of work, variety of professional experience, knowledge of foreign languages, and digital and informatics skills. The survey was submitted to all PSD employees. Additionally, all participants were asked to submit their updated Curriculum Vitae in conformity to the latest European versions of the survey compilation.(c)Accurate training of the nursing professionals regarding the Synergy Model: an accredited training course was established inside PSD and was open to all nursing personnel (head nurses and others) operating in the Cardiac Surgery Unit.(d)Structural changes to the Operating Unit: management of beds, assessment of the complexity of patients, and identification of the skills related to nursing staff have been enclosed in a procedure.(e)Development of job descriptions and monitoring records and the promotion of nursing competencies: These tools were created by nurses’ groups for different clinical areas (surgical, medical, critical services, and training) and represent Operating Instructions active in the IRCCS Policlinico San Donato. These are useful for describing nursing activities and monitoring the advancement of skills required by care pathways.

## 6. Limitations, Perspectives, and Future Implications

The current discursive paper has several limitations, mainly related to the limited capacity of generalization that the contents of these scientific communication approaches might have. Therefore, the contents of this article should be intended to boost scientific debate and generate hypotheses for future robust studies. In the described experience, some relevant information required to precisely assess the changes within the organization is lacking due to the informative purpose of the paper. Furthermore, some associations between the implementation of the Synergy Model and patient-reported outcomes, such as health-related quality of life, should be assessed in future investigations.

The implementation of the Synergy Model was developed to test its features in a real clinical context and determine critical issues or aspects requiring change before the future implementation in all hospital Operative Units. In this sense, the Cardiac Surgery Unit was chosen as a pilot unit for the Synergy Model’s implementation, as it matched the real criteria that would have promoted a cultural change in a well-established organization, such as the willingness of the head nurse, nursing, and support staff members [37]. A team-based organizational nursing delivery model [38] was enhanced to ensure an ecosystem for supporting the transition of patients towards rehabilitation and primary care services. The team leader and his group deliver assistance to specific patients in a determined work shift and provide global assistance to the patient. This deals with the functional model-related fragmentation of the care and provides a discharge report to the staff of rehabilitation and primary care services. In this way, nursing care delivery at the Cardiac Surgery Unit took the first real step towards patient centralization by no longer using the task’s ad functions model [39].

## 7. Conclusions

The rationale for implementing the Synergy Model to facilitate the transition from in-hospital to primary care seems promising. However, future evaluations of clinical and patient-reported outcomes have to longitudinally evaluate the effects of the created ecosystem on the safe and effective transition of patients once discharged, towards rehabilitation after cardiac surgery, and from rehabilitation to primary care services. This experience might generate a more robust rationale for implementing system-level strategies to create an ecosystem to sustain a sound transition across different levels of care. Stronger evidence is required to assess whether the Synergy Model implementation improves the transition from in-hospital to primary care, and this discursive paper intends to generate hypotheses and robust future projects.

## Data Availability

The data presented in this study are available on request from the corresponding author.

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
