# Peer review of "Does the Synergy Model Implementation Improve the Transition from In-Hospital to Primary Care? The Experience from an Italian Cardiac Surgery Unit, Perspectives, and Future Implications"

_ijerph, 2022, doi:10.3390/ijerph19095624_

Round 1
Reviewer 1 Report
Dellafiore et al. presented the experience from an Italian Cardiac Surgery Unit for implementation of the Synergy Model to improve the transition from in-hospital to primary care.
The paper explained clearly the AACN Synergy model for patient centered care, and how it is adapted to cardiac surgery unit through two phases, which are analysis of perceived barriers and organizational improvements.
What are lacking, however, are data and details that can help audience utilize the experience in the cardiac surgery unit for other applications of the synergy model. It will be helpful to identify what leads to the eight under-themes and three main themes for the hindrances, and if possible, include case studies. Also, for the organization changes inside PSD, it will be helpful to outline the reasons behind these changes, so readers can learn what leads to the decision behind these changes.
Author Response
Dear Editor and reviewers,
Thank you for the opportunity to review our manuscript for consideration in the IJERPH. We took into account all the points suggested in order to improve the quality of the paper. Please find below point-by-point answers to the recommendations. Changes are highlighted in red in the track version of the manuscript

Reviewer 2 Report
This work describes the experience of the authors regarding the adaptation of the Synergy Model in a Cardiac Surgery Unit and for guiding a smooth transition of patients toward rehabilitation and primary care services.
This study did not report how the implementation of the model improves the quality of life of patients.
The clarity of this work could be improved if the authors organised this work in a conventional way (introduction, methods, results, discussion…)
Line 154
- “The first phase of the project was to examine and interpret the barriers perceived by the nursing staff as an obstacle for the following Synergy Model implementation, thorough a mixed sequential method study [36]. This phase was conducted between 2015 and 2016 and consisted of semi-structured interviews following a grid of 5 pre-defined questions to the nursing staff”
Please detail the methodology used, including the selection and number of nurses included; how the content analysis was done; How the themes emerged.
- Given the results coming from the interviews, it was possible to provide a questionnaire to 117 nurses in order to define perceived hindrances from a quantitative approach.
Please provide the questionnaire. Who designed and validated the questionnaire?
Author Response

(The authors gave the same response as above.)

Round 2
Reviewer 1 Report
The authors have improved the paper with additional information, and also added limitations to the paper. The paper can be accepted with minor changes in language. Specifically, on line 155, what does 'felt' mean? It is not immediately clear for me. Thanks!
Author Response
thank you

Reviewer 2 Report
thank you for the responses.
Author Response
thank you
FD
